# A FTIR and DFT Combination Study to Reveal the Mechanism of Eliminating the Azeotropy in Ethyl Propionate–n-Propanol System with Ionic Liquid Entrainer

**DOI:** 10.3390/ijms241310597

**Published:** 2023-06-25

**Authors:** Yan-Zhen Zheng, Rui Zhao, Yu-Cang Zhang, Yu Zhou

**Affiliations:** 1College of Ocean Food and Biological Engineering, Jimei University, Xiamen 361021, China; yanzhenzheng@jmu.edu.cn; 2School of Chemistry and Chemical Engineering, Qingdao University, Qingdao 266071, China; jiangyuxinqdu@126.com

**Keywords:** solution chemistry, excess spectroscopy, interaction, azeotropy, density-functional theory

## Abstract

Ionic liquids (ILs) have presented excellent behaviors in the separation of azeotropes in extractive distillation. However, the intrinsic molecular nature of ILs in the separation of azeotropic systems is not clear. In this paper, Fourier-transform infrared spectroscopy (FTIR) and theoretical calculations were applied to screen the microstructures of ethyl propionate–n-propanol–1-ethyl-3-methylimidzolium acetate ([EMIM][OAC]) systems before and after azeotropy breaking. A detailed vibrational analysis was carried out on the *v*(C=O) region of ethyl propionate and *v*(O–D) region of n-propanol-*d*_1_. Different species, including multiple sizes of propanol and ethyl propionate self-aggregators, ethyl propionate–n-propanol interaction complexes, and different IL–n-propanol interaction complexes, were identified using excess spectroscopy and confirmed with theoretical calculations. Their changes in relative amounts were also observed. The hydrogen bond between n-propanol and ethyl propionate/[EMIM][OAC] was detected, and the interaction properties were also revealed. Overall, the intrinsic molecular nature of the azeotropy breaking was clear. First, the interactions between [EMIM][OAC] and n-propanol were stronger than those between [EMIM][OAC] and ethyl propionate, which influenced the relative volatilities of the two components in the system. Second, the interactions between n-propanol and [EMIM][OAC] were stronger than those between n-propanol and ethyl propionate. Hence, adding [EMIM][OAC] could break apart the ethyl propionate–n-propanol complex (causing the azeotropy in the studied system). When *x*([EMIM][OAC]) was lower than 0.04, the azeotropy still existed mainly because the low IL could not destroy the whole ethyl propionate–n-propanol interaction complex. At *x*(IL) > 0.04, the whole ethyl propionate–n-propanol complex was destroyed, and the azeotropy disappeared.

## 1. Introduction

Ethyl propionate, a widely used chemical reagent in various industries such as flavors, pharmaceuticals, and organic synthesis [1,2,3], and n-propanol, a versatile fuel additive and solvent, are important components produced during liquor production [4]. The separation of ethyl propionate and n-propanol is crucial for maintaining the quality and flavor of liquor and enabling their reuse. However, these two compounds form an azeotrope at atmospheric pressure, making their separation challenging using conventional distillation methods and requiring specialized techniques. Several distillation methods, including azeotropic distillation [5], extractive distillation [6,7], and pressure-swing distillation [8,9], have been developed to separate azeotropes. Among these methods, extractive distillation is commonly employed due to its high efficiency. It involves introducing a third component, known as an entrainer, into the system to modify the relative volatility of the two components and break the azeotropy [10]. The choice of entrainer plays a critical role in the effectiveness of extractive distillation. Traditionally, inorganic salts and organic solvents such as dimethyl sulfoxide (DMSO) and ethylene glycol (EG) have been used as entrainers in extractive distillation [11,12]. However, these conventional entrainers have limitations, such as environmental pollution, low selectivity, and pipeline blockage. Therefore, there is an urgent need to explore economically viable, environmentally friendly, and efficient entrainers for extractive distillation.

In recent years, ILs, a large class of salts that remain liquid at approximately room temperature, have been regarded as superior entrainers ascribed to their characteristic qualities: negligible vapor pressures, excellent dissolving capacities, structure and property designability, high stabilities, low melting points, and so on [13,14,15,16,17,18,19]. ILs have been used as entrainers in the separation of various azeotropic systems, for example, alcohol–water/ester/ketone aliphatic/halogenated hydrocarbons [13,14,15,16], water–alcohol/tetrahydrofuran [17,18], and aromatic–aliphatic hydrocarbons [19]. In the separation of alcohol and ethyl propionate azeotropic systems, acetate-anion-based (1-alkyl-3-methylimidazolium acetate) and halogen-anion-based (1-alkyl-3-methylimidazolium chloride) ILs show excellent separation performance [20,21,22]. For example, in an ethyl propionate and n-propanol azeotropic system, [EMIM][OAC] exhibits a better separation performance than the conventional entrainer EG [22]. Only a small amount of the IL (*x*(IL) = 0.04) is needed to separate the azeotropic system. In addition, [EMIM][OAC] is nonvolatile, which makes the IL easy to recycle and reuse. These features also greatly reduce the total annual cost of separation and protect the environment [22]. Thus, ILs have drawn increasing attention in extractive distillation.

The macroscopic behaviors of ILs in azeotrope separation are due to the microstructures of the system and interactions between the ILs and the azeotropic compounds. Importantly, increasing the use of ILs and their optimization to improve their use as entrainers are also dependent on understanding the microstructures of and micro-interactions between ILs and the other components. The microstructures and interaction properties of liquid systems are fundamental research topics in the scientific community, and those of IL-containing solvent mixtures, including IL–ethanol, IL–water, IL–acetone, IL–acetonitrile, and IL–DMSO systems, have been widely studied [23,24,25,26,27,28,29,30,31,32,33,34]. For example, Chen et al. performed experimental (FTIR) and theoretical (DFT) analyses to provide insights into the molecular interactions between 1-methoxyethyl-3-methylimidazolium bis(trifluoromethylsulfonyl)imide and three polar solvents. An ion cluster/ion pair/cation–cosolvent was identified in the work, and the interactions between the IL and DMSO were the strongest [23]. Roth et al. performed DFT and FTIR experiments to illustrate the hydrogen-bond interactions in a 1-ethyl-3-methylimidazolium bis(trifluoromethylsulfonyl)imide-methanol system. They found that interactions occurred between the anion and the methanol hydroxyl group. Ion pairs demonstrating a neutral complex in an environment analogous to pure methanol were found when *x*(IL) was lower than 0.1 [27]. Köddermann et al. performed DFT and FTIR experiments on the interaction between an IL and water and found that due to the different interaction strengths between the IL and water, the vibrational region in the hydroxyl group was a good indicator to evaluate the polarity of the IL [29].

The above works have greatly enriched our perception of the interactions between ILs and organic cosolvents. However, a large proportion of the results in the literature are based on IL–cosolvent binary systems. In extractive distillation, an IL is added as a third component into the azeotropic system, and the mixtures are composed of at least three components. The microstructural features of IL–organic solvent systems could assist in understanding the volatility modifications and phase equilibrium modifications of the systems. However, the structural properties of trinary mixtures containing an IL and azeotrope cannot be completely revealed, which makes the nature of azeotropy breaking with an IL unclear. Hence, more work needs to be carried out on the microstructural features of trinary systems. In this investigation, the microstructures of an azeotrope (ethyl propionate–propanol) and [EMIM][OAC] trinary systems were analyzed via FTIR and theoretical calculations (the DFT method). In this study, [EMIM][OAC] was selected as the representative IL because of its capability to disrupt azeotrope formation in an ethyl propionate–propanol system [22]. Excess spectroscopy [35,36] was performed on the original infrared spectra to disclose the micro-interaction-associated transformation in the mixture and identify the interaction. The microstructures and the related interaction properties of the systems before and after azeotropy elimination were carefully studied to further reveal the intrinsic cause of the azeotropy breaking in the ethyl propionate–n-propanol system with the applied IL entrainer.

## 2. Results

### 2.1. Infrared Spectra of Pure Ethyl Propionate, n-Propanol-d_1_, and IL

The microstructural features of the azeotrope–[EMIM][OAC] trinary system were complicated. Different sizes of IL, n-propanol, and ethyl propionate self-aggregations and various interaction complexes between different components, for instance, IL–ethyl propionate, IL–n-propanol, and ethyl propionate–n-propanol, may have been present in the mixtures. It is difficult to reveal the microstructure of an n-propanol–ethyl propionate–[EMIM][OAC] system by directly investigating the infrared spectral features of this system. Furthermore, to illustrate the removal of azeotropy with an IL, the structural properties of the azeotropic system should also be understood. Thus, the features of the infrared spectra of the n-propanol and ethyl propionate in the n-propanol–ethyl propionate and IL–azeotrope systems were specifically focused on to reach the above goals. To eliminate the influence of the *v*(C–H) vibration of the IL in the vibrational region of the hydroxyl group of n-propanol, n-propanol-*d*_1_ was applied instead of n-propanol in the FTIR experiments. The ATR-FTIR spectra of pure n-propanol-*d*_1_, ethyl propionate, and [EMIM][OAC] are shown in Figure 1. It was difficult to reveal the structural features from the overlapped infrared region. In Figure 1, we can observe that the 1600 cm^−1^–2800 cm^−1^ range does not overlap. The band observed in the 2050 cm^−1^ to 2700 cm^−1^ region corresponds to the *v*(O–D) stretching vibration of n-propanol-*d*_1_. Ethyl propionate and IL do not exhibit obvious absorptions in this region. The 1650 cm^−1^–1800 cm^−1^ range belongs to the *v*(C=O) of ethyl propionate. N-propanol-*d*_1_ and IL do not have obvious absorptions in this region. Special attention was given to these two regions, and the subsequent analysis primarily centered around them.

According to the literature [22], when *x*([EMIM[OAC]) is larger than 0.04, the azeotropy of the ethyl propionate–n-propanol-*d*_1_ system is destroyed. In this study, two ethyl propionate–n-propanol-*d*_1_–[EMIM][OAC] systems with *x*(IL) fixed at 0.02 and 0.2 were selected to study the microstructural property changes in the azeotropic systems before and after eliminating azeotropy, which can further reveal the intrinsic molecular cause of the eliminated azeotropy. Before investigating the two trinary systems, the IR spectral features of the corresponding binary systems, ethyl propionate–[EMIM][OAC], [EMIM][OAC]–n-propanol-*d*_1_, and ethyl propionate–n-propanol-*d*_1_, were also analyzed to help understand the microstructures of the trinary systems.

### 2.2. The IR Spectral Features of the v(C=O) Region

C=O is the functional group of ethyl propionate. It can reflect microenvironmental changes in ethyl propionate. In this work, the *v*(C=O) region was analyzed first. It is centered in approximately the 1800 cm^–1^–1650 cm^−1^ range and at approximately 1736.9 cm^−1^ in pure ethyl propionate. The spectral features of the ethyl propionate–n-propanol-*d*_1_/[EMIM][OAC] systems were investigated first. However, due to the insolubility of ethyl propionate and [EMIM][OAC], the spectral features of this system were not obtained. Thus, the investigated IL may not interact with ethyl propionate.

Figure 2A1 presents the FTIR spectra of the ethyl propionate–n-propanol-*d*_1_ system in the *v*(C=O) region, ranging from 1800 cm^−1^ to 1650 cm^−1^, with a decrement of 0.1 and *x*(ethyl propionate) ≈ 1.0–0.1. As the n-propanol-*d*_1_ concentration gradually increases, the intensity of the band observed at 1736.9 cm^−1^ in pure ethyl propionate gradually weakens, accompanied by a blueshift to 1740.8 cm^−1^ in the *x*(ethyl propionate) = 0.1 mixture. Additionally, a new band emerges at approximately 1720.2 cm^−1^, which gradually redshifts with the increasing n-propanol-*d*_1_ concentration. This band is absent in pure ethyl propionate, and its intensity increases with the increasing n-propanol-*d*_1_ concentration. Notably, when *x*(ethyl propionate) is reduced to 0.1, the relative strength of this new band exceeds that of the band at 1736.9 cm^−1^ present in pure ethyl propionate. These observations suggest a possible interaction between ethyl propionate and n-propanol.

The excess spectra, which are obtained by subtracting the ideal spectra from the actual spectra, provide insights into the species changes within the mixture [23,26,33,34,35,36]. Figure 2A2 presents the excess spectra of the n-propanol-*d*_1_-ethyl propionate system. In this figure, each concentration exhibits one positive peak and one negative peak. Upon careful analysis, it was observed that the peak positions in Figure 2A2 remain relatively consistent across different concentrations. Specifically, the negative peak consistently appears at 1735.8 cm^−1^, while the positive band is consistently observed at approximately 1719.4 cm^−1^. In the context of excess spectra [23,26,33,34,35,36], the negative band corresponds to a decrease in the relative concentration of the species absorbed at the peak compared with the pure liquid. On the other hand, the positive band indicates an increase in the species absorbed at this wavenumber compared with the pure liquid. Therefore, the negative peak at approximately 1735.8 cm^−1^ suggests a decrease in the corresponding species compared with pure ethyl propionate, while the positive peak at 1719.4 cm^−1^ indicates an increase in the corresponding species relative to pure ethyl propionate.

Figure 2B1,C1 depicts the infrared spectra of the n-propanol-*d*_1_-ethyl propionate-IL systems. In the *x*(IL) = 0.2 system, non-uniform mixing was observed at higher concentrations of ethyl propionate, leading us to analyze only the mixtures in a uniform state. When *x*(IL) is 0.02, the spectral characteristics of this system resemble those of the n-propanol-*d*_1_-ethyl propionate system. Specifically, as the concentration of n-propanol-*d*_1_ gradually decreases, the intensity of the peak at 1736.9 cm^−1^ in pure ethyl propionate weakens while exhibiting a blueshift (from 1736.9 cm^−1^ in neat n-propanol-*d*_1_ to 1740.8 cm^−1^ at *x*(n-propanol-*d*_1_) = 0.1). Additionally, a new band emerges at approximately 1720.2 cm^−1^, which gradually redshifts with the increasing n-propanol-*d*_1_ concentration. This band is absent in pure ethyl propionate, and its intensity increases with the increasing n-propanol-*d*_1_ concentration. When *x*(ethyl propionate) is reduced to 0.1, its relative strength becomes almost equal to the band at 1736.9 cm^−1^ present in pure ethyl propionate. In the ethyl propionate-n-propanol-*d*_1_-IL (*x*(IL) = 0.2) system, the spectral features of *v*(C=O) significantly differ from those of the ethyl propionate-n-propanol-*d*_1_ system and the ethyl propionate-n-propanol-*d*_1_–IL (*x*(IL) = 0.02) system. No new band in the lower wavenumber range was observed in this system. The peak at 1736.9 cm^−1^ in pure ethyl propionate gradually shifts to lower wavenumbers, accompanied by a decrease in intensity as the concentration of n-propanol-*d*_1_ increases. These results suggest substantial differences in the microstructure of the n-propanol-*d*_1_–ethyl propionate–[EMIM][OAC] (*x*(IL) = 0.2) system compared with those of the n-propanol-*d*_1_-ethyl propionate and n-propanol-*d*_1_-ethyl propionate-[EMIM][OAC] (*x*(IL) = 0.02) systems.

The excess spectra of the two trinary systems are depicted in Figure 2B2,C2. The characteristics of the excess spectra in the trinary systems resemble those of the ethyl propionate-n-propanol-*d*_1_ system. Each excess spectrum exhibits a positive peak at a lower wavenumber and a negative peak at a higher wavenumber. Through careful analysis, we observed that the peak wavenumbers of the excess peaks remained consistent for each system at the studied concentrations. In the trinary systems with the lower IL concentration (*x*(IL) = 0.02), the negative peak and positive peak were fixed at approximately 1736.0 cm^−1^ and 1719.3 cm^−1^, respectively. In the trinary system with the higher IL concentration (*x*(IL) = 0.2), these peaks were centered at approximately 1736.3 cm^−1^ and 1725.5 cm^−1^, respectively. By considering the results for the three systems, we found that the wavenumbers of the negative peaks exhibited minimal variation. The slight difference may be attributed to the solvent effect. The negative peak was associated with a decrease in the species present in pure ethyl propionate, which was possibly related to the absorptions of ethyl propionate self-aggregations. The peak positions of the positive excess peaks in the ethyl propionate-n-propanol-*d*_1_ system and the trinary system with lower IL concentrations showed minimal variation. This indicates that the species related to the positive excess peaks in these two systems are the same, namely, the n-propanol–ethyl propionate interaction complex. In the n-propanol-*d*_1_-ethyl propionate-[EMIM][OAC] (*x*(IL) = 0.2) system, the positive excess peak (1725.5 cm^−1^) appears at a higher wavenumber compared with that (1719.3 cm^−1^) in the n-propanol-*d*_1_–ethyl propionate and n-propanol-*d*_1_–ethyl propionate–[EMIM][OAC] (*x*(IL) = 0.02) systems. This peak may not be associated with the ethyl propionate–n-propanol interaction complex. Additionally, it does not relate to the ethyl propionate–IL interaction complex due to the relatively weak interactions between ethyl propionate and the IL, as they do not mix with each other. This discrepancy may be related to the ethyl propionate monomer or other-sized self-aggregates.

### 2.3. The Assignment of Excess Peaks in the v(C=O) Region

In the n-propanol-*d*_1_–ethyl propionate system and the two n-propanol-*d*_1_–ethyl propionate–IL systems, we observed peaks at approximately 1719 cm^−1^ and 1736/1725.5 cm^−1^, which can be attributed to the n-propanol–ethyl propionate interaction complex and various ethyl propionate aggregators, respectively. To validate our inference and assign the excess peaks accurately, we employed density-functional theory (DFT) calculations, which are commonly used in spectral analysis for species assignment [23,26,33,34,35,36]. In this study, we optimized the structures of the n-propanol–ethyl propionate interaction complex, ethyl propionate, and different ethyl propionate aggregators to assign the excess peaks observed in Figure 2A2,B2,C2. The results of these calculations are presented in Table 1.

Table 1 reveals that the *v*(C=O) stretching frequency of the ethyl propionate monomer exhibited the highest wavenumber, which gradually decreased with an increase in the number of monomers in the aggregators. The *v*(C=O) stretching frequency of the ethyl propionate–n-propanol interaction complex was the lowest, confirming its association with the peak at 1719 cm^−1^. Based on the calculated *v*(C=O) stretching frequencies of different ethyl propionate self-aggregators, we determined that the negative excess peak observed at approximately 1736 cm^−1^ in Figure 2A2,B2 was associated with smaller ethyl propionate aggregators and monomers. Conversely, the positive excess peak at approximately 1725 cm^−1^ in Figure 2C2 was related to larger ethyl propionate aggregators. In the pure system, ethyl propionate predominantly existed as smaller aggregators and monomers. However, upon the addition of n-propanol, the smaller aggregators and monomers diminished due to the formation of the ethyl propionate–n-propanol interaction complex. In the trinary system with lower IL concentrations, there was still an excess of n-propanol available to interact with ethyl propionate, resulting in the detection of peaks associated with the ethyl propionate–n-propanol interaction complex. Conversely, in the system with higher IL concentration, all the n-propanol-*d*_1_ molecules favored interaction with the IL, leading to the compression of ethyl propionate molecules. Consequently, ethyl propionate formed larger aggregators, and the peaks related to these larger ethyl propionate aggregators are observed in Figure 2C2.

### 2.4. The Infrared Spectra Feature of the v(O–D) Region

Similar to Section 2.2, we initially analyzed the spectral characteristics of the binary systems (n-propanol-*d*_1_-ethyl propionate and n-propanol-*d*_1_-[EMIM][OAC]) in the *v*(O–D) region. In the case of ethyl propionate-n-propanol-*d*_1_, the entire concentration range was investigated. However, for the n-propanol-*d*_1_–IL system, only mixtures with higher n-propanol-*d*_1_ concentrations were analyzed to eliminate the azeotropy, as a mole fraction of 0.04 of IL was required. The infrared spectra of the n-propanol-*d*_1_-ethyl propionate and [EMIM][OAC] –n-propanol-*d*_1_ systems in the 2700 cm^−1^ to 2070 cm^−1^ region are presented in Figure 3A1,B1, respectively. In the 2700 cm^−1^ to 2070 cm^−1^ region, the *v*(O–D) stretching frequency of pure n-propanol-*d*_1_ was centered at approximately 2471.8 cm^−1^. The *v*(O–D) stretching frequency exhibited distinct changes in the two binary systems. In the ethyl propionate-n-propanol-*d*_1_ system, as the concentration of n-propanol-*d*_1_ decreased (from top to bottom), the intensity of *v*(O–D) monotonically decreased. The peak at 2471.8 cm^−1^, observed in pure n-propanol-*d*_1_, gradually shifted to higher wavenumbers. A total of 2 shoulder bands at higher wavenumbers than 2471.8 cm^−1^ were detected. Moving from higher to lower wavenumbers, the two peaks were designated as peak 2 and peak 1. Peak 1 appeared before peak 2, and its intensity was greater than that of peak 2 until *x*(n-propanol-*d*_1_) > 0.4. It was also stronger than the peak at 2471.8 cm^–1^ when *x*(propanol-*d*_1_) = 0.4. The relative intensity of peak 2 was the highest for *x*(n-propanol-*d*_1_) < 0.4. In the n-propanol-*d*_1_–IL system (Figure 3B1), the relative intensity of *v*(O–D) gradually weakened and shifted to lower wavenumbers. The bands became broader as [EMIM][OAC] concentration increased. A new band at a lower wavenumber emerged, and its relative intensity gradually strengthened. The relative intensity of the bands at 2471.8 cm^−1^ in pure n-propanol-*d*_1_ progressively decreased.

The excess spectra of the n-propanol-*d*_1_–ethyl propionate and n-propanol-*d*_1_-IL systems are depicted in Figure 3A2,B2, respectively. Each excess spectrum displayed distinct peaks, indicating different species changes and interactions between the components, as reported in the literature [23,26,33,34,35,36]. In both binary systems, a negative band in the 2500 cm^−1^ to 2400 cm^−1^ range was observed across all studied concentration ranges. This broad band consisted of multiple peaks, suggesting the involvement of multiple species changes. The shift towards higher wavenumbers indicated a decrease in the size of alcohol aggregations. The literature suggests that absorptions in this wavenumber range primarily originate from larger alcohol aggregations such as cyclic tetramers, pentamers, and larger structures [37,38,39]. The negative intensity in this region indicates a reduction in the larger alcohol aggregations in the n-propanol-*d*_1_–ethyl propionate and [EMIM][OAC]-n-propanol-*d*_1_ mixtures compared with pure n-propanol-*d*_1_ [23,26,33,34,35,36]. It is known that larger aggregations correspond to lower wavenumbers for *v*(O–D). The blueshift of this band indicated a transition from larger to smaller aggregations [37,38,39]. In the ethyl propionate-n-propanol-*d*_1_ system, the negative band gradually shifted from 2420.7 cm^−1^ (*x*(n-propanol-*d*_1_) = 0.9) to 2467.8 cm^−1^ (x(n-propanol-*d*_1_) = 0.1). Conversely, in the n-propanol-*d*_1_–IL system, the shift occurred from 2475.7 cm^−1^ (*x*(n-propanol-*d*_1_) = 0.9) to 2483.9 cm^-1^ (*x*(n-propanol-*d*_1_) = 0.5). These results indicate that IL more effectively disrupts n-propanol aggregations compared with ethyl propionate.

Apart from the broad negative bands, the excess spectra of the binary systems exhibited distinct characteristics. In the ethyl propionate–n-propanol-*d*_1_ system, a broad positive excess band (at approximately 2688 cm^−1^ to 2500 cm^−1^) with a peak position higher than that of the negative band was observed. Conversely, the n-propanol-*d*_1_–[EMIM][OAC] system displayed the opposite behavior, with the positive band appearing at a lower wavenumber than the negative band. It is evident that the positive peaks in both binary systems consist of multiple peaks, indicating complex species interactions. (If a positive peak consists of only one peak, it would typically be a sharp peak with a relatively constant position throughout the concentration range. However, our results demonstrate the opposite behavior). In the ethyl propionate–n-propanol-*d*_1_ system, the positive excess band (Figure 3A2) was composed of two peaks. The wavenumbers of the two peaks can be readily obtained at the lowest concentration of n-propanol-*d*_1_ (*x*(n-propanol-*d*_1_) = 0.1) and the highest concentration of n-propanol-*d*_1_ (*x*(n-propanol-*d*_1_) = 0.9). They were 2623.6 cm^−1^ and 2586.5 cm^−1^, respectively. The relative strength of the excess peaks in the lower wavenumber (2586.5 cm^−1^) gradually weakened, while that at the higher wavenumber showed the opposite behavior. It was larger than that at the higher wavenumber when *x*(n-propanol-*d*_1_) was higher than 0.4. For *x*(n-propanol-*d*_1_) < 0.4, it was the reverse. Following the literature [37,38,39], the peak at the lower wavenumber (approximately 2586.5 cm^−1^) was related to the absorption of alcohol cyclic trimers. Its positive property indicated the increased magnitude of the trimer in the mixture compared with pure n-propanol-*d*_1_ [23,26,33,34,35,36]. Its relative amount continuously decreased with the reduction in n-propanol-*d*_1_. The higher wavenumber peak (approximately 2623.6 cm^−1^), whose intensity gradually increased with the increase in ethyl propionate, appeared with the addition of ethyl propionate. Clearly, this peak was from the interaction complex between n-propanol-*d*_1_ and ethyl propionate. The existence of the ethyl propionate–n-propanol-*d*_1_ interaction complex was also the cause of the azeotropy in this system. In the n-propanol-*d*_1_–[EMIM][OAC] system, the broad positive band also consisted of multiple peaks. Its wavenumber gradually shifted from 2310.8 cm^−1^ to 2254.4 cm^−1^ with the reduction in n-propanol-*d*_1_. A small positive peak at approximately 2670 cm^−1^ was also detected. These positive peaks also appeared upon adding the IL. They must be related to the interaction of [EMIM][OAc] with n-propanol-*d*_1_. In the n-propanol-*d*_1_–[EMIM][OAC] system, we also observed a band related to the alcohol cyclic trimer. This band was only present in the excess spectra at *x*(n-propanol-*d*_1_) = 0.9 and 0.8. This result also indicates that the relative magnitude of the cyclic alcohol trimer was larger than that of pure n-propanol-*d*_1_ in the n-propanol-*d*_1_–IL system at *x*(propanol-*d*_1_) = 0.9 and 0.8. However, it decreased when the alcohol content continuously increased. Due to the solvent effect, its wavenumber was slightly lower than that in the ethyl propionate–n-propanol-*d*_1_ system. Based on this result, we can also determine that n-propanol-*d*_1_ was more influenced by the IL than by ethyl propionate.

The infrared and excess spectra of *v*(O–D) in the two ethyl propionate-n-propanol-*d*_1_-IL systems are presented in Figure 3C1,D1,C2,D2, respectively. The infrared spectral characteristics of these two systems exhibited significant differences. The ethyl propionate-n-propanol-*d*_1_-IL *x*(IL) = 0.02 system (Figure 3C1,C2) showed similar spectral characteristics to the ethyl propionate-n-propanol-*d*_1_ system (Figure 3A1,A2). Specifically, the infrared spectra of *v*(O–D) at 2471.8 cm^−1^ observed in pure n-propanol-*d*_1_ gradually shifted to higher wavenumbers. A total of 2 shoulder bands were visible at higher wavenumbers near 2471.8 cm^−1^. On the other hand, the spectral characteristics of the ethyl propionate-n-propanol-*d*_1_-IL *x*(IL) = 0.2 system resembled those of the n-propanol-*d*_1_-IL system. The bands at 2471.8 cm^−1^ in pure propanol progressively shifted to lower wavenumbers (Figure 3D1). The *v*(O–D) bands became broader with the increasing ethyl propionate concentration. A new band emerged at a lower wavenumber, and its relative intensity gradually strengthened. Regarding the excess spectra of the 2 n-propanol-*d*_1_-ethyl propionate-IL systems, similar to the n-propanol-*d*_1_-ethyl propionate and IL-n-propanol-*d*_1_ systems, a broad negative band was observed in each excess spectrum in the 2500–2400 cm^−1^ range. This negative band was associated with a decrease in the presence of larger alcohol aggregators in the mixture. In the ethyl propionate–n-propanol-*d*_1_–IL (*x*(IL) = 0.02) system, similar to the ethyl propionate–n-propanol-*d*_1_ system, a broad positive band consisting of 2 peaks at approximately 2587.7 cm^−1^ (related to the propanol cyclic trimer) and 2623.5 cm^−1^ (related to the ethyl propionate–propanol interaction complex) was observed. The relative intensities of these two peaks show similar variation trends as the ethyl propionate–n-propanol-*d*_1_ system. Except for these characteristics, in this system, a small broad band in the lower wavenumber of the negative band was also detected after careful analysis. Its intensity was relatively low and gradually decreased. This result was similar to the band arising from the IL–n-propanol-*d*_1_ interaction complex. Except for the nonexistence of the band related to the alcohol cyclic trimer, the excess spectra of ethyl propionate–n-propanol-*d*_1_–IL (*x*(IL) = 0.2) also had a broad positive band and a small positive peak at approximately 2670 cm^−1^ related to the IL–n-propanol interaction complexes at the lower wavenumber of the negative band. The wavenumber of the broad band shifted from 2305.8 cm^−1^ to 2258.3 cm^−1^ when *x*(n-propanol-*d*_1_) was reduced from 0.7 to 0.3.

### 2.5. The Assignment of the Excess Peaks in the v(O–D) Region

The 2400–2500 cm^−1^ region and the peak at approximately 2586 cm^−1^ in the *v*(O–D) region, as mentioned in the literature [37,38,39], are associated with larger alcohol aggregators and cyclic trimers, respectively. In the excess spectra of the n-propanol-*d*_1_-ethyl propionate and n-propanol-*d*_1_-ethyl propionate-IL (*x*(IL) = 0.02) systems, a peak appeared at approximately 2623.6 cm^−1^ upon the addition of ethyl propionate. This peak is attributed to the interaction between ethyl propionate and n-propanol-d1. Additionally, in the excess spectra of the n-propanol-*d*_1_-IL and ethyl propionate-n-propanol-d1-IL (*x*(IL) = 0.2) systems, a peak below 2311 cm^−1^ and a small peak at approximately 2670 cm^−1^ were observed upon the addition of IL. These peaks were likely due to the formation of n-propanol–IL interaction complexes. To verify our assignments and gain further insight into the structural changes in n-propanol in the mixture, DFT calculations were performed in this study, which complemented the experimental data [23,26,33,34,38]. The most stable geometries and corresponding frequencies were optimized for the interaction between n-propanol and ethyl propionate, as well as various n-propanol–IL complexes. Furthermore, to support our assignments, the most stable geometries of the alcohol trimer and larger aggregators (such as cyclic tetramers and pentamers) were also calculated. In IL–cosolvent mixtures, the ILs can exist in the form of ion clusters of different sizes, ion pairs, and individual ions. Based on previous studies [40,41], we considered two [EMIM]^+^−two [OAC]––n-propanol complexes (2[EMIM]^+^–2[OAc]^−^–CH_3_CH_2_CH_2_OD), one [EMIM]^+^−one [OAC]^−^–propanol complex ([EMIM]^+^–[OAc]^−^–CH_3_CH_2_CH_2_OD), and one [EMIM]^+^/[OAC]^−^–propanol complex ([EMIM]^+^/[OAc]^−^–CH_3_CH_2_CH_2_OD) to represent the interaction complexes between ion clusters and n-propanol, ion pairs and n-propanol, and ions and n-propanol, respectively.

The optimized *v*(O–D) frequencies for n-propanol-d1 in different aggregators are summarized in Table 2. The calculated frequencies follow the sequence: 2[EMIM]^+^–2[OAC]^−^–CH_3_CH_2_CH_2_OD < [EMIM]^+^–2[OAC]^−^–CH_3_CH_2_CH_2_OD < [OAC]^−^–CH_3_CH_2_CH_2_OD < pentamer of CH_3_CH_2_CH_2_OD < tetramer of CH_3_CH_2_CH_2_OD < trimer of CH_3_CH_2_CH_2_OD < ethyl propionate–CH3CH_2_CH_2_OD < [EMIM]^+^–CH_3_CH_2_CH_2_OD. Consistent with the literature [37,38,39], the larger the alcohol aggregator, the lower the vibrational frequency of the most intense *v*(O–D) mode. Our calculated results confirm this observation, thus further validating the accuracy of our calculations. With the exception of [EMIM]^+^–CH_3_CH_2_CH_2_OD, the frequency of *v*(O–D) in the IL–n-propanol interaction complex was lower than those in different n-propanol self-aggregators. The *v*(O–D) frequency in the cation–alcohol complex was the highest among the interaction complexes. Moreover, the calculated intensity of this peak was relatively weak compared with the other interaction complexes. Therefore, the small positive peak at approximately 2670 cm^−1^ can be assigned to the absorption of the cation–n-propanol interaction complex. The positive peaks below 2311 cm^−1^ originated from the anion–n-propanol, ion cluster–n-propanol, and ion pair–n-propanol interactions. The calculated *v*(O–D) frequency in the ethyl propionate–CH_3_CH_2_CH_2_OD complex was higher than that in the n-propanol self-aggregators, providing confirmation of the presence of the positive excess band at approximately 2623.6 cm^−1^, which arose from the ethyl propionate–propanol interaction complex.

### 2.6. The Interaction Properties between [EMIM][OAC]/Ethyl Propionate and n-Propanol

To illustrate the interaction properties between ethyl propionate/IL and n-propanol, the most stable geometries of the identified species, including the ion cluster–n-propanol, ion pair–n-propanol, cation/anion–n-propanol, and ethyl propionate–n-propanol interaction complexes, were specifically focused on and are presented in Figure 4. In solvent mixtures, hydrogen bonds are some of the most important interactions between different components. In this work, hydrogen-bond interactions were specifically focused on. The hydrogen bonds between different constituents in the optimized geometries were identified first via geometrical rules: H∙∙∙O distances smaller than 2.5 Å (the van der Waals atomic radii of the hydrogen and oxygen atoms) [42]. They are labeled with dashed lines for each complex in Figure 4. The identified hydrogen bonds were further confirmed via the topological criterion: the existence of the bond critical point (BCP) at the hydrogen bond and the topological properties in the conventional range (0.002 au to 0.040 au for electron density (*ρ*_BCP_) and 0.02 au to 0.15 au for the Laplacian of electron density (∇^2^*ρ*_BCP_)) [43]. Specifically, the hydrogen bonds between n-propanol and IL/ethyl propionate were investigated. Their distances were labeled next to the corresponding hydrogen bonds. The corresponding topological properties of these hydrogen bonds are listed in Table 3. It is clear that hydrogen bonds were found between n-propanol and IL/ethyl propionate. In the ion cluster–n-propanol, ion pair–n-propanol, and ethyl propionate–n-propanol interaction complexes, both the hydrogen atom and the oxygen atom of the n-propanol hydroxyl group formed hydrogen bonds with the IL or ethyl propionate. In the anion/cation–n-propanol interaction complex, the hydroxyl group of n-propanol participated in hydrogen bonding. Thus, the hydroxyl group was the interaction site of n-propanol. For different hydrogen bonds in one complex, the shorter the hydrogen bond distance and the larger the ∇^2^*ρ*_BCP_ are, the stronger the hydrogen bond is [44]. Thus, the hydrogen bond related to the hydrogen atom in the hydroxyl group was the strongest. For ethyl propionate, the hydrogen bond related to the oxygen atom in C=O was stronger. This result further confirms the important role of the hydroxyl group of n-propanol in hydrogen bond formation and illustrates that the C=O group of ethyl propionate was the main interaction group. Thus, the vibrational characteristics of the C=O group of ethyl propionate and the hydroxyl group of n-propanol could be used to detect changes in these systems.

In addition to identifying the formation and relative strength of the hydrogen bond, the topological properties of ∇^2^*ρ*_BCP_ and energy density (*H*_BCP_) could also help to reveal the hydrogen bond properties. In the atoms in molecules (AIM) theory [44,45], when both ∇^2^*ρ*_BCP_ and *H*_BCP_ are less than zero, the property of the corresponding hydrogen bond is strong, shared, and covalently dominant. If ∇^2^*ρ*_BCP_ is larger than zero and *H*_BCP_ is less than zero, the investigated hydrogen bond is medium-strength, closed-shell, and covalently dominant. In another case wherein both topological properties were positive, weak-strength, closed-shell, and electrostatically dominant interactions were found for the corresponding hydrogen bonds. In Table 3, for the hydrogen bonds between the anion and the hydroxyl group, ∇^2^*ρ*_BCP_ was larger than zero, while *H*_BCP_ was less than zero. Thus, these hydrogen bonds were medium-strength, closed-shell, and covalently dominant interactions. For the other hydrogen bonds, both ∇^2^*ρ*_BCP_ and *H*_BCP_ were positive. Therefore, the hydrogen bonds between ethyl propionate and n-propanol and those between the cation and n-propanol were weak-strength, closed-shell, and electrostatically dominant interactions.

The interaction energies between IL/ethyl propionate and n-propanol are labeled below the corresponding optimized interaction complexes. It was clear that the interaction energies between IL and n-propanol were much larger than those between ethyl propionate and n-propanol. Thus, the IL can break the ethyl propionate–n-propanol interaction complexes.

## 3. Discussion

In the mixing process, [EMIM][OAC] could not mix with ethyl propionate, while it could be mixed with n-propanol-*d*_1_ in any proportion. Thus, the interactions between [EMIM][OAC] and n-propanol were stronger than those between [EMIM][OAC] and ethyl propionate. The interaction strength difference between IL/ethyl propionate and n-propanol influenced the relative volatility of the two components in the system, which was one of the reasons for breaking the azeotropy. In the other case, the interactions between n-propanol and [EMIM][OAC] were stronger than those between n-propanol and ethyl propionate. Hence, adding [EMIM][OAC] could break apart the ethyl propionate–n-propanol interaction complex (the cause of the azeotropy in the ethyl propionate-n-propanol system). When *x*([EMIM][OAC]) was lower than 0.04, the azeotropy still existed mainly because the low IL could not completely destroy the whole ethyl propionate–n-propanol interaction complex. This result was illustrated in the ethyl propionate–n-propanol–IL system when *x*(IL) = 0.02, wherein the ethyl propionate–n-propanol interaction complex was also detected in the *v*(C=O) and *v*(O–D) regions. At *x*(IL) > 0.04 (*x*(IL) = 0.2 was applied in this work), the entire ethyl propionate–n-propanol complex was destroyed, and we could not observe the ethyl propionate–n-propanol interaction complex. The above two cases together resulted in the elimination of the azeotropy.

## 4. Materials and Methods

### 4.1. Chemicals

The chemicals in this work, deuterated n-propanol (CH_3_CH_2_CH_2_OD, 99%), ethyl propionate (99%), and [EMIM][OAC] (99%), were obtained from Cambridge Isotopes Laboratories, Sinopharm Chemical Reagent Co., Ltd. (Shanghai, China), and Cheng Jie Chemical Co. Ltd. (Shanghai, China), respectively. Water in the air is easily absorbed by ILs and affects the structural properties of the binary and trinary mixtures. To remove water, the IL [EMIM][OAC] was placed under reduced pressure at T = 313.15 K for a minimum of 24 h before use.

### 4.2. FTIR Spectroscopy

The FTIR spectra of all the samples were collected with a Nicolet 5700 FTIR spectrometer equipped with different attenuated total reflectance (ATR) units (ZnSe and Ge) at room temperature (~296 K). The spectra collected with the ZnSe ATR unit (with a 2.43 refractive index, 45° angle of incidence, and 12 reflections) were applied for the analysis of the *v*(O–D) region of n-propanol-*d*_1_. Those collected with the Ge ATR unit (with 4 refractive indices, a 60° angle of incidence, and 7 reflections) were used for the analysis of the *v*(C=O) region of ethyl propionate. Each spectrum ranged from 650 cm^−1^ to 4000 cm^−1^ with a 2 cm^−1^ resolution, 32 parallel scans, and a 0-filling factor. The excess spectroscopy procedure devised by Yu’s group [35,36] was performed for the spectra of the mixtures to enhance the resolution of the IR spectra and obtain the absorption peaks related to different associations in the systems.

### 4.3. DFT Calculations

The DFT calculations related to the geometries, energies, and frequencies of the monomers, different self-aggregators, and different interaction complexes were carried out using the M06–2X functional with the standard 6–311g(d,p) basis set implemented in the Gaussian 16 software package [46]. All the geometric configuration optimization processes were performed without any symmetric constraints. The optimized energy minimum configurations were confirmed without imaginary frequency. Based on the geometry optimization, the interaction energies between different constituents were obtained via the M06-2X/6-311+g(2d,p) method with counterpoise (CP) correction [47]. To better understand the nature of the hydrogen-bond interactions between n-propanol and ethyl propionate/IL, the AIM theory was conducted on the most stable interaction geometries of IL–n-propanol and ethyl propionate–n-propanol [44]. The search for the BCP and the corresponding topological descriptors at the BCP was performed via the Multiwfn 3.5 suite [48]. Three typical topological descriptors related to the hydrogen-bond instances and properties, namely, *ρ*_BCP_, ∇^2^*ρ*_BCP_, and *H*_BCP_, were obtained in the AIM analysis.

## 5. Conclusions

In summary, FTIR and DFT calculations were employed with the aim of understanding the microstructural properties of ethyl propionate–n-propanol–[EMIM][OAC] systems and further illustrating the intrinsic molecular properties of the IL applied as an entrainer in extractive distillation. A detailed vibrational analysis was carried out on the *v*(C=O) region of ethyl propionate and the *v*(O–D) region of n-propanol-*d*_1_. The spectral features of the ethyl propionate–n-propanol-*d*_1_–IL systems before and after the azeotropy breaking were understood based on the results for the ethyl propionate–n-propanol-*d*_1_ and ethyl propionate/n-propanol-*d*_1_–IL binary systems.

In the n-propanol systems, the decrease in n-propanol was accompanied by a reduction in the large sizes of propanol self-aggregations (larger than a tetramer). The relative amount of the n-propanol trimer in the binary systems and ethyl propionate–n-propanol–IL (*x*(IL) = 0.02) first increased and then decreased with the reduction in n-propanol. In contrast, the reduction in ethyl propionate was accompanied by a reduction in smaller aggregators in the ethyl propionate systems. The ethyl propionate–n-propanol interaction complex was identified in the ethyl propionate-n-propanol-*d*_1_ and ethyl propionate–n-propanol-*d*_1_–IL (*x*(IL) = 0.02) systems. They were the cause of the azeotropy phenomenon in the ethyl propionate–n-propanol system and ethyl propionate–n-propanol–IL systems at lower IL concentrations (*x*(IL) < 0.04). Different IL–n-propanol interaction complexes (ion–propanol, ion pair–n-propanol, and ion cluster–n-propanol) were identified in the n-propanol-*d*_1_–IL and ethyl propionate–n-propanol-*d*_1_–IL (*x*(IL) = 0.2) systems. The interaction properties were analyzed for the identified interaction complexes. The hydrogen bonds between the IL and n-propanol are stronger than those between ethyl propionate and n-propanol. Those between the anion of IL and the hydroxyl group are medium-strength, closed-shell, and covalently dominant interactions. Those between ethyl propionate and n-propanol and those between the cation of IL and n-propanol are weak-strength, closed-shell, and electrostatically dominant interactions.

Based on the above conclusions, the intrinsic molecular properties of azeotropy breaking were revealed. First, the interactions between [EMIM][OAC] and n-propanol were stronger than those between [EMIM][OAC] and ethyl propionate, which influenced the relative volatilities of the two components in the system. Second, the interactions between n-propanol and [EMIM][OAC] were stronger than those between n-propanol and ethyl propionate. Hence, adding [EMIM][OAC] could break apart the ethyl propionate–n-propanol interaction complex (the cause of the azeotropy in the ethyl propionate–n-propanol system). When *x*([EMIM][OAC]) was lower than 0.04, the azeotropy still existed mainly because the low IL could not completely destroy the whole ethyl propionate–propanol interaction complex. At *x*(IL) > 0.04, the entire ethyl propionate–propanol complex was destroyed, and the azeotropy disappeared.

## Figures and Tables

**Figure 1 ijms-24-10597-f001:**
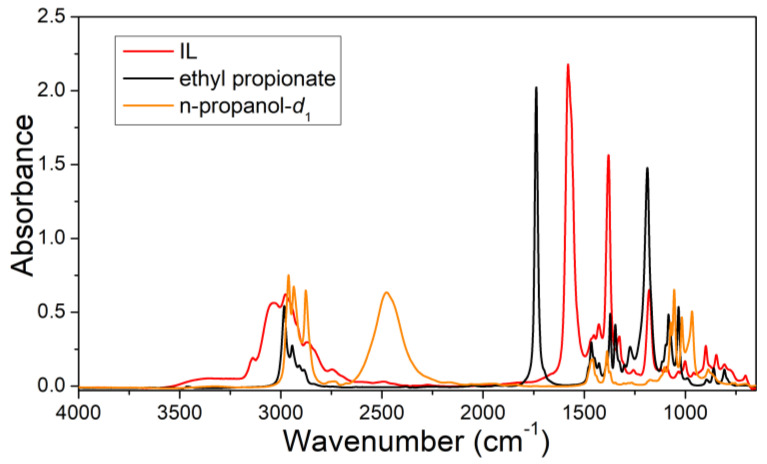
ATR-FTIR spectra of pure IL [EMIM][OAC], ethyl propionate, and deuterated n-propanol (CH_3_CH_2_CH_2_OD) in the range of 4000–650 cm^−1^.

**Figure 2 ijms-24-10597-f002:**
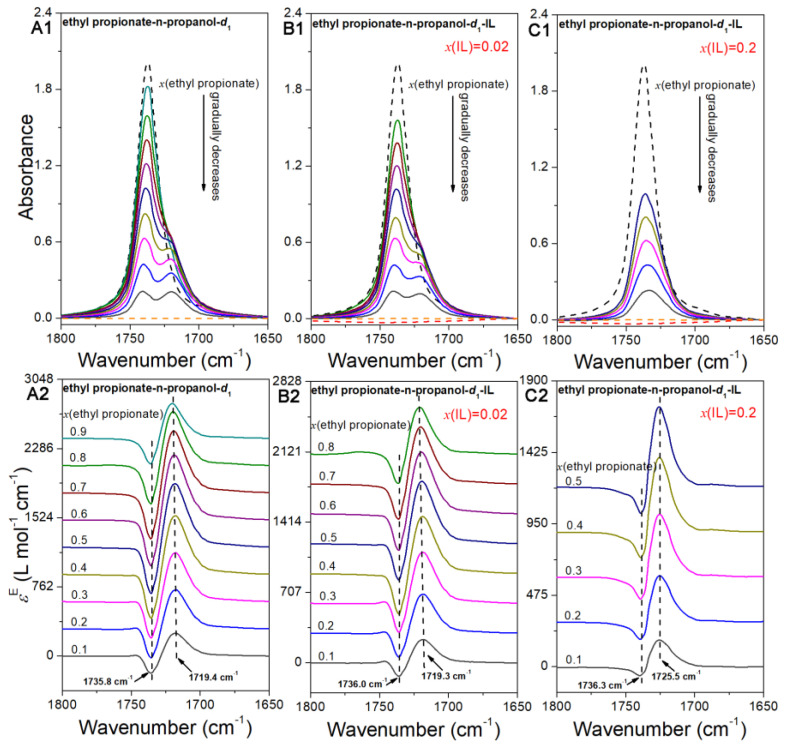
IR (**A1**,**B1**,**C1**) and excess (**A2**,**B2**,**C2**) spectra of the ethyl propionate–n-propanol-*d*_1_ and ethyl propionate–n-propanol-*d*_1_–[EMIM][OAC] systems in the 1800 cm^−1^–1650 cm^−1^ region. In (**A2**,**B2**,**C2**), the excess spectra are elevated equally to see the peaks more clearly. The black dashed line is the IR spectrum of pure ethyl propionate, and those in orange and red are the IR spectra of pure deuterated n-propanol (CH_3_CH_2_CH_2_OD) and [EMIM][OAC], respectively. The other colorful lines are the IR spectra (**A1**,**B1**,**C1**) and the corresponding excess spectra (**A2**,**B2**,**C2**) of the mixtures.

**Figure 3 ijms-24-10597-f003:**
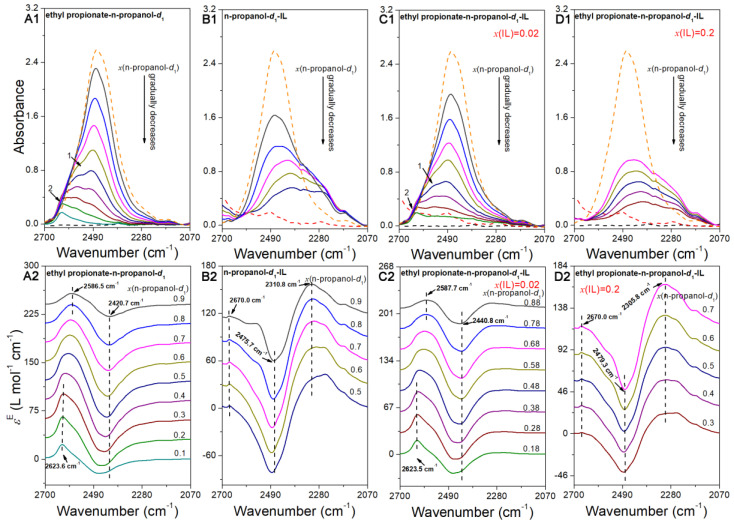
IR (**A1**,**B1**,**C1**,**D1**) and excess (**A2**,**B2**,**C2**,**D2**) spectra of the ethyl propionate–n-propanol-*d*_1_, n-propanol-*d*_1_–IL and ethyl propionate–n-propanol-*d*_1_–[EMIM][OAC] systems at 2700 cm^−1^–2070 cm^−1^. In (**A2**,**B2**,**C2**), the excess spectra are elevated equally to see the peaks more clearly. The dashed orange line is the IR spectrum of pure deuterated n-propanol (CH_3_CH_2_CH_2_OD), and those in black and red are the IR spectra of pure ethyl propionate and [EMIM][OAC], respectively. The other colorful lines are the IR spectra (**A1**,**B1**,**C1**,**D1**) and the corresponding excess spectra (**A2**,**B2**,**C2**,**D2**) of the mixtures. Number 1 and 2 are the identified peaks in the IR spectra of A1.

**Figure 4 ijms-24-10597-f004:**
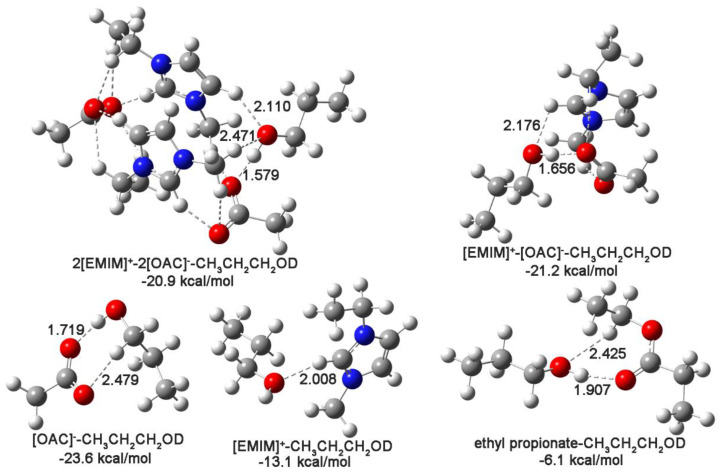
The optimized geometries of ion cluster–n-propanol (2[EMIM]^+^–2[OAC]^−^–CH_3_CH_2_CH_2_OD), ion pair–n-propanol ([EMIM]^+^–[OAC]^−^–CH_3_CH_2_CH_2_OD), anion–n-propanol ([OAC]^−^–CH_3_CH_2_CH_2_OD), cation–n-propanol ([EMIM]^+^–CH_3_CH_2_CH2OD), and ethyl propionate–n-propanol interaction complexes. The color balls in red, blue, light grey and dark grey are the oxygen, nitrogen, hydrogen and carbon atoms, respectively.

**Table 1 ijms-24-10597-t001:** Calculated results of *v*(C=O) in the possible interaction complexes related to ethyl propionate.

Complex	Calculated Frequency	Calculated Intensity
Ethyl propionate monomer	1863.3	221.4
Ethyl propionate dimer	1855.2	214.5
	1854.7	230.8
Ethyl propionate trimer	1849.5	442.2
	1845.1	184.7
	1834.3	156.7
Ethyl propionate–CH_3_CH_2_CH_2_OD	1830.3	320.3

**Table 2 ijms-24-10597-t002:** Calculated results for *v*(O–D) in the possible interaction complexes related to n-propanol. The data in bold are the most intense *v*(O–D) modes in the most stable optimized complexes of n-propanol self-aggregators.

Complex	Calculated Frequency	Calculated Intensity	Observed Frequency
[EMIM]^+^–CH_3_CH_2_CH_2_OD	2833.4	50.1	~2670.0
Ethyl propionate–CH_3_CH_2_CH_2_OD	2768.5	277.8	~2623.6
Cyclic trimer of CH_3_CH_2_CH_2_OD	**2712.8**	**452.5**	~2586.5
	2697.1	371.3	
	2660.0	45.7	
Cyclic tetramer of CH_3_CH_2_CH_2_OD	2572.1	233.0	~2486.1–2420.7
	**2540.9**	**1111.4**	
	**2540.2**	**1113.4**	
	2466.2	0.0	
Cyclic pentamer of CH_3_CH_2_CH_2_OD	2571.6	267.4	
	2538.4	163.6	
	**2509.1**	**1167.2**	
	**2492.2**	**1715.0**	
	2433.2	98.3	
[OAC]^−^–CH_3_CH_2_CH_2_OD	2442.2	554.7	~2310.1–2254.4
[EMIM]^+^–[OAC]^−^–CH_3_CH_2_CH_2_OD	2297.8	751.2	
2[EMIM]^+^–2[OAC]^−^–CH_3_CH_2_CH_2_OD	2232.8	688.7	

**Table 3 ijms-24-10597-t003:** Some topological properties in a.u at the BCPs of the hydrogen bonds between n-propanol and ethyl propionate/IL in the studied complexes obtained using the AIM theory. The atoms in bold are the atoms in n-propanol.

Interaction Complex	Hydrogen Bond	Hydrogen Bond Distance (Å)	*ρ* _BCP_	∇^2^*ρ*_BCP_	*H* _BCP_
[EMIM]^+^–CH_3_CH_2_CH_2_OD	C2–H∙∙∙**O**	2.008	0.022	0.091	0.003
Ethyl propionate–CH_3_CH_2_CH_2_OD	C=O∙∙∙**H–O**	1.907	0.025	0.104	0.003
	C–H∙∙∙**O**	2.425	0.011	0.042	0.002
[OAC]^−^–CH_3_CH_2_CH_2_OD	O∙∙∙H–**O**	1.719	0.043	0.138	−0.003
	O∙∙∙**H–C**	2.479	0.012	0.037	0.001
[EMIM]^+^–[OAC]^−^–CH_3_CH_2_CH_2_OD	O∙∙∙**H–O**	1.656	0.038	0.143	−0.008
	C2–H∙∙∙**O**	2.176	0.018	0.077	0.003
2[EMIM]^+^–2[OAC]^−^–CH_3_CH_2_CH_2_OD	O∙∙∙**H–O**	1.579	0.040	0.150	−0.013
	C4–H∙∙∙**O**	2.110	0.020	0.073	0.002
	C3–H∙∙∙**O**	2.471	0.009	0.030	0.001

## Data Availability

All relevant data are within this paper.

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
