# Peer review of "A FTIR and DFT Combination Study to Reveal the Mechanism of Eliminating the Azeotropy in Ethyl Propionate–n-Propanol System with Ionic Liquid Entrainer"

_ijms, 2023, doi:10.3390/ijms241310597_

Round 1

Reviewer 1 Report

The manuscript concerns the analysis of the microstructures of n-propanol, ethyl propionate and 1-ethyl-3-methylimidzolium acetate mixtures. Unfortunately, not all conclusions in the manuscript are supported by experimental evidence.
The authors consider the system of n-propanol-d1 with IL and EP, and not n-propanol with additives. N-propanol-d1 differs in physical properties - it has a different mass and boiling point than n-propanol. Deuterium does not significantly affect the energy of the particle, but it does affect the wavenumbers for individual oscillations and the Zero Point Vibrational Energy. This causes unobvious changes in the behavior of such a molecule in complex systems (ternary mixtures and conglomerates). There are also significant differences in the strength of the detuer bond relative to the hydrogen bond. It seems to me that you can't transfer the properties of n-propanol-d1 to n-propanol one-to-one.

The introduction is taken partly from doi.org/10.1016/j.molliq.2022.118492

Figure 1 is well captioned, while lines 118-119 state that the spectrum represents n-propanol. This is important because the band originating from the OH group is above 3100 cm-1, while the spectrum shows a stretching OD band at about 2300 cm-1.

Lines 126-127: repetition

Lines 219-221: "It also cannot interact with the ethyl propionate–IL interaction complex because ethyl propionate does not interact with the IL, as they cannot mix with each other. This may be related to the ethyl propionate monomer or other size self-aggregators." - It cannot be said that there is no interaction between the IL and the ester. The interactions are weaker and of a different type than between IL and alcohol - there is a predominance of cation-carbonyl dispersion interactions, which are described in the literature.

Lines 283, 308, 366, 394: "As from the literature, the absorptions in this wavenumber range were mainly from the larger aggregations of alcohol, for instance cyclic tetramers, pentamers and larger ones"; "As from the literature [37–39], the peak in the lower wavenumber (approximately 2586.5 cm–1) was related to the absorption of alcohol cyclic trimers."; "The 2400–2500 cm−1 region and the peak around 2586 cm–1 of the v(O‒D) region, as from the literature [37–39], were related to larger alcohol aggregators and cyclic trimers, respectively."; "According to the literature [37–39], the larger the alcohol aggregator is, the smaller the vibrational frequency of the most intense v(O‒D). Our calculated results confirmed this conclusion, which further verified the accuracy of our calculated results." - The authors often refer to references 37-39. In publications 37 and 39, I found no evidence for the statement made by the authors in the cited places. The effects described there are for water and methanol, not for higher alcohols. The authors of these publications state that the data for methanol do not apply to higher alcohols, for which there are different effects. The authors of publication 38 were the authors of this manuscript.

Line 300: "Clearly, the positive peaks in the two binary systems were composed of more than one peak." - Again, I see no basis for this statement.

It seems to me that in order for the manuscript to meet the standards of scientific publication, additional data are needed to support the theses made by the authors.

Author Response

Comment 1: The manuscript concerns the analysis of the microstructures of n-propanol, ethyl propionate and 1-ethyl-3-methylimidzolium acetate mixtures. Unfortunately, not all conclusions in the manuscript are supported by experimental evidence.

The authors consider the system of n-propanol-d1 with IL and EP, and not n-propanol with additives. N-propanol-d1 differs in physical properties - it has a different mass and boiling point than n-propanol. Deuterium does not significantly affect the energy of the particle, but it does affect the wavenumbers for individual oscillations and the Zero Point Vibrational Energy. This causes unobvious changes in the behavior of such a molecule in complex systems (ternary mixtures and conglomerates). There are also significant differences in the strength of the detuer bond relative to the hydrogen bond. It seems to me that you can't transfer the properties of n-propanol-d1 to n-propanol one-to-one.

Response: We appreciate your kind reminder and would like to clarify some aspects of our work. Our focus in this study was specifically on the vibrational region of the hydroxyl group of propanol. This region contains abundant information about the structural characteristics of the liquid under investigation.

In the FTIR experiment, we utilized n-propanol-d1 instead of n-propanol to eliminate the influence of the v(C−H) vibration of the IL. This is a commonly employed approach in FTIR experiments, where deuterated molecules are used as substitutes for non-deuterated ones to mitigate spectral overlap and interference.

To address concerns about the differences between n-propanol and n-propanol-d1, we conducted several analyses. Firstly, we compared the boiling points of n-propanol (97°C), n-propanol-d1 (97.4°C), and ethyl propionate (99.1°C). From the engineering perspective, these slight variations in boiling points are not expected to significantly impact the system behavior or the formation of azeotropes.

Additionally, we performed geometry optimizations of different n-propanol aggregators and the interaction complexes between ILs/ethyl propionate and n-propanol. Our findings indicate that the most stable geometries of these aggregators and complexes do not exhibit substantial differences when using n-propanol-d1 instead of n-propanol. The hydrogen-bond distances and interaction energies also remain similar. Besides, the sequence of the calculated frequencies of v(O−H) in different complexes is the same as those of v(O−D).

Based on these investigations, we maintain that the experimental evidence could support the conclusions presented in the manuscript. We acknowledge your concerns and appreciate the opportunity to clarify these aspects of our work.

Comment 2: The introduction is taken partly from doi.org/10.1016/j.molliq.2022.118492

Response: Thanks for your kind reminder. We have revised the introduction section in the revised manuscript (pages 1 to 2, lines 36-53).

Comment 3: Figure 1 is well captioned, while lines 118-119 state that the spectrum represents n-propanol. This is important because the band originating from the OH group is above 3100 cm-1, while the spectrum shows a stretching OD band at about 2300 cm-1.

Response: Thanks for your careful reading and kind reminder. We have corrected the errors throughout the revised manuscript.

Comment 4: Lines 126-127: repetition

Response: Thanks for the reviewer’s careful reading. We have deleted the repetition in the revised manuscript (page 3, lines 130-132).

Comment 5: Lines 219-221: "It also cannot interact with the ethyl propionate–IL interaction complex because ethyl propionate does not interact with the IL, as they cannot mix with each other. This may be related to the ethyl propionate monomer or other size self-aggregators." - It cannot be said that there is no interaction between the IL and the ester. The interactions are weaker and of a different type than between IL and alcohol - there is a predominance of cation-carbonyl dispersion interactions, which are described in the literature.

Response: Thanks a lot for your professional guidance. We have corrected the errors in the revised manuscript (page 6, lines 227 to 229).

Comment 6: Lines 283, 308, 366, 394: "As from the literature, the absorptions in this wavenumber range were mainly from the larger aggregations of alcohol, for instance cyclic tetramers, pentamers and larger ones"; "As from the literature [37–39], the peak in the lower wavenumber (approximately 2586.5 cm–1) was related to the absorption of alcohol cyclic trimers."; "The 2400–2500 cm−1 region and the peak around 2586 cm–1 of the v(O‒D) region, as from the literature [37–39], were related to larger alcohol aggregators and cyclic trimers, respectively."; "According to the literature [37–39], the larger the alcohol aggregator is, the smaller the vibrational frequency of the most intense v(O‒D). Our calculated results confirmed this conclusion, which further verified the accuracy of our calculated results." - The authors often refer to references 37-39. In publications 37 and 39, I found no evidence for the statement made by the authors in the cited places. The effects described there are for water and methanol, not for higher alcohols. The authors of these publications state that the data for methanol do not apply to higher alcohols, for which there are different effects. The authors of publication 38 were the authors of this manuscript.

Response: Thank you for your thoughtful consideration and polite reminder. The existing literatures predominantly focus on examining the aggregation behavior of water and methanol. Unfortunately, we were unable to find any relevant articles that specifically discuss the aggregation forms of n-propanol in solution. Nonetheless, studying the aggregation behavior of methanol and water can still provide valuable insights that can guide our research effectively.

Comment 7: Line 300: "Clearly, the positive peaks in the two binary systems were composed of more than one peak." - Again, I see no basis for this statement.

Response: We sincerely apologize for the confusion caused. If a positive peak consists of only one peak, it would typically be a sharp peak with a relatively constant position throughout the concentration range (we have also added these points in the revised manuscript (page 8, lines 320 to 323)). However, our results demonstrate the opposite behavior. For example, in Figure 3A2, the positive peak is not sharp but instead exhibits a gradual shift in position from a lower wavenumber to a higher wavenumber as the concentration of ethyl propionate increases. The wavenumbers of the two peaks can be easily determined at the lowest concentration of n-propanol (x(n-propanol) = 0.1) and the highest concentration of n-propanol (x(n-propanol) = 0.9). Specifically, they were measured to be 2623.6 cm‒1 and 2586.5 cm‒1, respectively.

Comment 8: It seems to me that in order for the manuscript to meet the standards of scientific publication, additional data are needed to support the theses made by the authors.

Response: Thank you for the reviewer's suggestions, which have provided us with valuable guidance. In future work, we will conduct more research and gather more data to provide stronger support for our conclusions.

Reviewer 2 Report

Several corrections should be made to the initial manuscript before being considered for publication.

1. The authors should explain the reason for choosing that particular IL ([emim][OAc])

2. The authors should explain clearly why they chose to use deuterated propanol for their IR studies.

3. The differences between the calculated and the measured IR frequencies is high; the authors should explain this.

4. The discussion of the IR results is very difficult to follow. This discussion should be shortened and clearly focused. Some plot of the band position (wavenumbers) vs x should be introduced for more clarity.

5. The following statement should be rephrased: "The band at 2050 cm–1–2700 cm–1 belongs the v(O–D) of n-propanol."

6. Fig. 1 should be re-scaled for x-scale by using increments of hundreds (e.g 2500, 2000, etc.cm-1).

7. The following paragraph should be rephrased to avoid repetition: "These two regions were specifically focused on, and the following analysis was mainly focused on these regions."

8. Which is section 3.2 mentioned on top of the page 7?

Minor corrections are necessary.

Author Response

Several corrections should be made to the initial manuscript before being considered for publication.

Comment 1. The authors should explain the reason for choosing that particular IL ([emim][OAc])

Response: Thank you for your helpful reminder. In the revised manuscript (page 3, lines 101 to 103), we have provided an explanation for selecting [EMIM][OAC] as the representative IL. This choice was made based on its capability to break the azeotrope of the ethyl propionate-propanol system.

Comment 2. The authors should explain clearly why they chose to use deuterated propanol for their IR studies.

Response: Thanks for your kind reminder. We chose to use deuterated propanol for our IR studies due to eliminate the influence of the v(C−H) of the IL in the v(O−H) (revised manuscript, page 3, lines 120 to 122).

Comment 3. The differences between the calculated and the measured IR frequencies is high; the authors should explain this.

Response: Thanks for your kind reminder. The differences between the calculated infrared frequencies in Gaussian and the experimental frequencies can be attributed to several factors. Some possible reasons include.

Approximation methods: Infrared calculations often employ approximation methods such as Density Functional Theory (DFT) or Hartree-Fock (HF). These methods may not accurately describe the vibrational properties of molecules, resulting in disparities between the calculated and experimental frequencies.

Dynamical effects: Calculations typically assume that molecules vibrate at static equilibrium, while in experiments, molecules can be influenced by dynamic factors such as temperature and pressure. These dynamical effects can contribute to the disparities between calculated and experimental frequencies.

Solvent effects: Calculations often neglect the influence of solvents or the surrounding environment on molecular vibrations. However, in experimental settings, molecules may exist in solutions or interact with other molecules, leading to differences between calculated and experimental frequencies.

It should be noted that disparities between calculated and experimental frequencies do not necessarily imply that the calculations are incorrect. Calculated frequencies can provide valuable information about molecular vibrational behavior when compared with experimental results.

Comment 4. The discussion of the IR results is very difficult to follow. This discussion should be shortened and clearly focused. Some plot of the band position (wavenumbers) vs x should be introduced for more clarity.

Response: We apologize for the confusion caused by the lengthy and unclear discussion of the IR results. We have revised the discussions of the IR results in the revised manuscript (pages 4 to 11, lines 153 to 426).

Comment 5. The following statement should be rephrased: "The band at 2050 cm–1–2700 cm–1 belongs the v(O–D) of n-propanol."

Response: Thanks for your kind reminder. We have rephrased the sentences in the revised manuscript (page 3, lines 125 to 127).

Comment 6. Fig. 1 should be re-scaled for x-scale by using increments of hundreds (e.g 2500, 2000, etc.cm-1).

Response: Thanks for your valuable suggestions. We have re-scaled for x-scale by using increments of hundreds in Figure 1.

Comment 7. The following paragraph should be rephrased to avoid repetition: "These two regions were specifically focused on, and the following analysis was mainly focused on these regions."

Response: Thanks for your careful reading. We have corrected the errors in the revised manuscript (page 3, lines 130 to 132).

Comment 8. Which is section 3.2 mentioned on top of the page 7?

Response: Thanks for your careful reading and kind reminder. It is section 2.2. We have corrected the errors in the revised manuscript (page 7, line 265).

Round 2

Reviewer 1 Report

The changes introduced by the Authors made the manuscript more transparent and precise. I found no further methodological errors in the revised manuscript. The text may still need to be edited in terms of language errors.

Reviewer 2 Report

The authors addressed all the comments raised by the reviewers. The manuscript can be accepted for publication.

Only few minor corrections should be made.